# Tricellulin, α-Catenin and Microfibrillar-Associated Protein 5 Exhibit Concomitantly Altered Immunosignals along with Vascular, Extracellular and Cytoskeletal Elements after Experimental Focal Cerebral Ischemia

**DOI:** 10.3390/ijms241511893

**Published:** 2023-07-25

**Authors:** Corinna Höfling, Steffen Roßner, Bianca Flachmeyer, Martin Krueger, Wolfgang Härtig, Dominik Michalski

**Affiliations:** 1Paul Flechsig Institute for Brain Research, University of Leipzig, Liebigstr. 19, 04103 Leipzig, Germany; corinna.hoefling@medizin.uni-leipzig.de (C.H.); steffen.rossner@medizin.uni-leipzig.de (S.R.); wolfgang.haertig@medizin.uni-leipzig.de (W.H.); 2Institute of Anatomy, University of Leipzig, Liebigstr. 13, 04103 Leipzig, Germany; bianca.flachmeyer@medizin.uni-leipzig.de (B.F.); martin.krueger@medizin.uni-leipzig.de (M.K.); 3Department of Neurology, University of Leipzig, Liebigstr. 20, 04103 Leipzig, Germany

**Keywords:** tricellulin, microfibrillar-associated protein 5, α-catenin, neurovascular unit, NVU, experimental stroke, focal cerebral ischemia

## Abstract

Along with initiatives to understand the pathophysiology of stroke in detail and to identify neuroprotective targets, cell-stabilizing elements have gained increasing attention. Although cell culture experiments have indicated that tricellulin, α-catenin and microfibrillar-associated protein 5 (MFAP5) contribute to cellular integrity, these elements have not yet been investigated in the ischemic brain. Applying immunofluorescence labeling, this study explored tricellulin, MFAP5 and α-catenin in non-ischemic and ischemic brain areas of mice (24, 4 h of ischemia) and rats (4 h of ischemia), along with collagen IV and fibronectin as vascular and extracellular matrix constituents and microtubule-associated protein 2 (MAP2) and neurofilament light chain (NF-L) as cytoskeletal elements. Immunosignals of tricellulin and notably MFAP5 partially appeared in a fiber-like pattern, and α-catenin appeared more in a dotted pattern. Regional associations with vascular and extracellular constituents were found for tricellulin and α-catenin, particularly in ischemic areas. Due to ischemia, signals of tricellulin, MFAP5 and α-catenin decreased concomitantly with MAP2 and NF-L, whereby MFAP5 provided the most sensitive reaction. For the first time, this study demonstrated ischemia-related alterations in tricellulin, MFAP5 and α-catenin along with the vasculature, extracellular matrix and cytoskeleton. Confirmatory studies are needed, also exploring their role in cellular integrity and the potential for neuroprotective approaches in stroke.

## 1. Introduction

Ischemic stroke is characterized by a critical reduction in blood supply for a brain area caused by acute vessel occlusion with various mechanisms, ultimately leading to tissue damage [1]. Treatment strategies currently focus on recanalizing occluded vessels via either intravenous thrombolysis or mechanical thrombectomy [2,3]. As these strategies are limited to a minority of patients and successful recanalization does not necessarily lead to favorable outcomes, efforts exist to develop neuroprotective substances serving as sole or additional treatments [4,5]. Research has therefore focused on a more detailed understanding of pathophysiological mechanisms, leading to long-lasting tissue damage following ischemia.

Regarding the cellular consequences of stroke, the neurovascular unit (NVU) has been established as a construct summarizing the local formation of neuronal, glial, and vascular elements [6,7,8]. Accordingly, experimental studies have started to explore NVU elements in the ischemic brain (e.g., [9]; overview in [10,11]). In addition to cellular impairments emerging from focal cerebral ischemia, the extracellular matrix, i.e., perineuronal nets (e.g., [12,13]) and fibronectin [14], has shown significant affections, at which fibronectin was found with a close regional association to the vasculature. Consequently, efforts have been made to identify structures serving as targets for neuroprotective approaches. In this context, cell-stabilizing elements have gained attention, as they likely have a crucial role during ischemia-related tissue damage [15,16]. As constituents of the cytoskeleton, microtubule-associated protein 2 (MAP2) and neurofilament light chain (NF-L) were found with a susceptible reaction in rodent brains subjected to focal cerebral ischemia [17,18,19]. Concomitantly observed alterations in cellular NVU constituents thereby support the assumption that a critical affection of cell-stabilizing elements, such as MAP2 and NF-L, represents a key mechanism for cellular degeneration due to ischemia and might be considered a neuroprotective target. 

A novel cell-stabilizing element critically impacting the NVU’s integrity might be tricellulin, a trans-membrane protein that has been found to realize the contact between three adjoining cells, seen in epithelial and non-epithelial cells [20,21,22,23]. In cell culture experiments, tricellulin was recently found to bind with α-catenin, thus forming a complex that is linked to cytoskeletal elements, such as actin [24]. As a unique feature, tricellulin seems to possess considerable mechanical resistance because tricellular contacts are occasionally characterized by high epithelial tension [25]. The assumption of highly dynamic processes at cell–cell contacts in conjunction with cytoskeletal changes is supported by an earlier cell culture-based study, indicating the ability of α-catenin to change its conformation depending on the force acting on the cytoskeleton [26]. 

From the variety of extracellular filaments potentially contributing to cellular integrity, the family of microfibrillar-associated proteins (MFAPs) has recently turned into focus. MFAPs represent extracellular matrix glycoproteins involved in processes such as assembling elastic microfibrils, remodeling and tissue homeostasis, which was particularly demonstrated for metabolic and cancer diseases [27]. MFAP5 in particular has been investigated in the adipose tissue of humans [28] and the infrapatellar fat pad in mice [29]. These studies have reported increased expression due to stimuli such as inflammation and high glucose levels. Further, MFAP5 was analyzed in human tissue samples of ovarian cancer, showing a higher mRNA expression in high-grade tumors [30], indicating a relationship between MFAP5 and rapid changes in matrix compositions. Remarkably, MFAP5 was found to be involved in the extracellular matrix remodeling of varicose veins [31] and might serve as a biomarker for cardiac remodeling in heart failure [32], leading to the presumption that MFAP5 may also play a role in cardiovascular diseases. 

However, tricellulin and MFAP5 have not yet been investigated in the non-ischemic and the ischemic brain along with elements of the NVU and the extracellular matrix. This study thus aimed to explore tricellulin, MFAP and α-catenin in rodent models of focal cerebral ischemia in conjunction with vascular and extracellular constituents, as well as the cytoskeletal elements MAP2 and NF-L. These insights might help to extend the perspective of the NVU and, especially, the knowledge on cell-stabilizing elements, which might serve as a target for neuroprotective strategies in stroke. 

## 2. Results

### 2.1. Cellular and Extracellular Consequences of Focal Cerebral Ischemia

In mice subjected to 24 h of focal cerebral ischemia, typical alterations in NVU elements were visible especially in the ischemic border zone of the neocortex, thus qualifying for an explorative analysis of cell-stabilizing elements in naive and ischemia-affected brain regions. In detail, slightly increased immunosignals of collagen IV and fibronectin were observed in the ischemic area, representing markers associated with the vasculature and the extracellular matrix (Figure 1A,B). These alterations were accompanied by a partially increased NF-L signal within the ischemic border zone and a markedly decreased MAP2 signal toward the ischemic area, representing cytoskeletal and thus cell-stabilizing elements that have already shown high vulnerability in the setting of ischemia. 

### 2.2. Immunosignals of Tricellulin, MFAP5 and α-Catenin in Naive and Ischemic Brain Regions

In brain tissues from mice subjected to 24 h of focal cerebral ischemia, tricellulin was detectable via immunofluorescence labeling, with a signal appearing diffusely arranged and, at least in part, in a fiber-like configuration in non-ischemic areas (right upper part in Figure 1C’’). Referring to the ischemic border zone of the neocortex, the immunosignal of tricellulin markedly decreased toward the area affected by ischemia. Particularly in the ischemic area, the tricellulin signal frequently appeared to be associated with vascular structures and extracellular matrix elements, as indicated by regionally overlapping signals from collagen IV and fibronectin (Figure 1C–C’’’). However, in the area covered by ischemia, single regions existed with a tricellulin signal comparable with non-affected areas, even though an overall decreased immunosignal of tricellulin was seen in ischemic areas.

Immunofluorescence-based detection of MFAP5 resulted in a signal comprising mainly fiber-like elements with a few regional accumulations in non-affected brain regions (right part in Figure 1D’’). In the ischemia-affected neocortex, the MFAP5 signal appeared decreased (Figure 1D’’) with a relatively sharp boundary at the border zone, as indicated by the changed collagen IV signal (Figure 1D). While merging the staining pattern, regional overlapping of MFAP5 with vascular and extracellular elements visualized by collagen IV and fibronectin was not visible (Figure 1D’’’).

Concerning α-catenin, histochemical detection revealed a diffusely arranged and a partially dotted signal in non-ischemic areas (right part in Figure 1E’’). In the ischemia-affected region, the signal appeared slightly diminished and, at least in part, in a more dotted pattern (Figure 1E’’). Concomitant labeling of collagen IV and fibronectin indicated a few accumulations of the dotted α-catenin signal regionally overlapping with vascular and extracellular elements.

### 2.3. Histochemical Patterns of Tricellulin, MFAP5 and α-Catenin Compared with MAP2 and NF-L in the Ischemia-Affected Neocortex and Subcortex

Immunosignals of tricellulin, MFAP5 and α-catenin were quantified along with signals from the cytoskeletal elements MAP2 and NF-L at the ischemic border zone of the neocortex and subcortex in brain tissues from mice subjected to 24 h of focal cerebral ischemia (Figure 2A). 

Compared with the contralateral, non-affected hemisphere, a gradual decrease in the tricellulin signal was observed along the neocortex toward the ischemia-affected area, accompanied by decreasing signals of MAP2 and NF-L, which was also found in the subcortex (Figure 2B). However, a statistically significant reduction in the tricellulin signal was evident only for the ischemia-affected subcortex (*p* = 0.028), whereas the performed inter-hemispheric comparison of signals covering the neocortical border zone (*p* = 0.360, *p* = 0.160) and the associated ischemic region of the neocortex (both *p* = 0.054) failed to reach statistical significance after careful consideration of performed multiple comparisons. Nevertheless, comparing the most medial area with the most lateral area covering the least affected and the most ischemic region, a significantly reduced tricellulin signal was detected in the most ischemic area (*p* = 0.012). An inter-hemispheric comparison of signals visualizing MAP2 and NF-L revealed a non-significantly increased signal for NF-L close to the border zone, followed by a significant decrease in both signals along the neocortex starting in the border zone and maintaining in the ischemic region (each *p* = 0.014). Further, significantly decreased signals of MAP2 and NF-L were found in the ischemic subcortex (each *p* = 0.014). Regarding the regional arrangement of tricellulin, MAP2 and NF-L, the merged immunofluorescence labeling indicated partial overlapping of fiber-like structures seen in both the non- and the ischemia-affected area (Figure 2B’,B’’).

For MFAP5, a gradual decrease in the immunosignal was observed along the ischemia-affected neocortex (Figure 2C). Thereby, the inter-hemispheric comparison yielded a significant reduction in the MFAP5 signal in the neocortex starting close to the ischemic border zone (*p* = 0.040) with an even more apparent reduction toward the region affected by ischemia (each *p* = 0.014). A significantly decreased MFAP5 signal was consistently found when comparing the most naive and most ischemia-affected areas (*p* = 0.012). Further, a significantly decreased MFAP5 signal was observed in the ischemic subcortex compared to the contralateral hemisphere (*p* = 0.014). Accompanied by the decreasing immunosignal of MFAP5 along the ischemia-affected neocortex, signals of MAP2 and NF-L decreased significantly (*p*-values ranging from 0.014 to 0.027), even though in this subset of analyses NF-L was found to decrease to a lesser degree. This observation was also evident for MAP2 and NF-L the ischemic subcortex (*p* = 0.014, *p* = 0.020). Concerning regional arrangements, MFAP5 seemed to overlap with a relevant part of fiber-like structures visualized by MAP2 und NF-L (Figure 2C’,C’’). Due to the decreased MAP2 signal in the ischemic-affected area, overlapping for MFAP5 and MAP2 appeared to be more prominent in the non-affected area.

The Immunofluorescence labeling of α-catenin also showed a gradually decreasing signal along the ischemia-affected neocortex (Figure 2D). Inter-hemispheric comparison revealed significant reductions in the α-catenin signal starting in the lateral border zone (*p* = 0.016) and maintaining toward the neocortical area affected by ischemia (each *p* = 0.014). In addition, the ischemic subcortex showed a significantly lowered α-catenin signal (*p* = 0.014). When comparing the most medial region with the most lateral region analyzed, a significantly decreased α-catenin signal was found in the most ischemic area (*p* = 0.012). Regarding MAP2 and NF-L, a non-significant increase in the NF-L signal was seen at the medial part of the border zone, whereas toward the ischemia-affected region, a reduction in both signals was observed (*p*-values ranging from 0.014 to 0.027), which was also evident for the ischemic subcortex (*p* = 0.014; *p* = 0.024). The regional arrangement of α-catenin with MAP2 and NF-L primarily appeared in a complementary manner, as α-catenin with its dotted appearance in the ischemia-affected area did not provide certain evidence of overlaps with both cytoskeletal elements (Figure 2D’,D’’).

### 2.4. Detection of Cell-Stabilizing Elements in Brains Subjected to Short-Term Focal Ischemia

To verify and extend the observations in mice after 24 h of focal cerebral ischemia, a subset of qualitative immunofluorescence-based analyses focused on tricellulin, MFAP5 and α-catenin in mice and rats subjected to 4 h of ischemia, respectively (Figure 3). 

In the neocortices of mice affected by 4 h of focal cerebral ischemia, a slightly decreased tricellulin immunosignal was visible in the ischemic area (Figure 3A’). Although the degree of reduction seemed to be lesser than that at 24 h after ischemia, the fiber-like configuration visible in non-ischemic areas (right part of Figure 3A’) mainly disappeared toward the ischemic area. The altered tricellulin signal was accompanied by a marked reduction in the MAP2 signal and a gradually increased NF-L signal directly at the ischemic border zone (Figure 3A). Regarding MFAP5, a markedly decreased immunosignal was found in the neocortical area covered by focal cerebral ischemia (Figure 3B’). Remarkably, the alteration in the MFAP5 immunosignal was accompanied by a comparable reduction in the MAP2 signal and a regional increase in the NF-L signal at the ischemic border zone of the neocortex (Figure 3B). In contrast to tricellulin and MFAP5, which partially appeared with a fiber-like configuration, the diffuse and partly dotted signal of α-catenin slightly decreased in neocortical areas that were relevantly affected after 4 h of focal cerebral ischemia (Figure 3C,C’).

In rats subjected to 4 h of focal cerebral ischemia, immunofluorescence labeling revealed comparable staining patterns of tricellulin, MFAP5 and α-catenin in the ischemia-affected neocortex as reported above for mice. In detail, the immunosignal of tricellulin was found to decrease in the ischemic area, and the fiber-like configuration became virtually lost (Figure 3D’). This alteration was accompanied by a notably decreasing MAP2 signal in the border zone of ischemia (Figure 3D). Regarding MFAP5, a decreased immunosignal was observed in the ischemia-affected neocortex, accompanied by a decreased MAP2 signal and a slightly increased appearance of NF-L (Figure 3E,E’). For α-catenin, a slightly decreased immunosignal in the neocortical area covered by ischemia became visible (Figure 3F’). Again, this alteration was accompanied by a marked decrease in the MAP2 signal, and NF-L was observed with a relatively weak signal gradually increasing in the border zone (Figure 3F). 

## 3. Discussion

This study aimed to explore tricellulin, MFAP5 and α-catenin in rodent models of focal cerebral ischemia in conjunction with vascular and extracellular components and the cytoskeletal elements MAP2 and NF-L. Efforts have been made to consider the well-established concept of the NVU with an adjacent extracellular matrix [6,7,14]. In addition, observations have been verified at another time point of ischemia and a further rodent species, as earlier studies have highlighted time-dependent pathophysiological mechanisms in stroke [1] and advances in the comparability between models [33,34]. 

To the authors’ best knowledge, this study showed, for the first time, tricellulin, MFAP5 and α-catenin along with NVU and cytoskeletal elements in the rodent brain subjected to focal cerebral ischemia. Thereby, immunosignals of tricellulin and MFAP5 mainly appeared in a diffusely arranged and partially fiber-like pattern with a marked impairment in ischemic regions, whereas α-catenin appeared more in a dotted pattern. 

Although an overall decreasing immunosignal of tricellulin was found due to focal cerebral ischemia, at least in part, a regional association was seen with the signal visualizing collagen IV as a representative of the basement membrane and thus the vascular component of the NVU [35]. This association appears plausible because the trans-membrane protein tricellulin is critically involved in maintaining contact between three cells at physiological barriers [22,36,37]. Although most available data on tricellulin are based on cell culture experiments involving epithelial cells (e.g., [20,24]), few studies have investigated its presence in conjunction with endothelial cells. In detail, tricellulin was discovered in cell culture- and immunofluorescence-based studies involving the intestines and pancreas of mice [38], capillary endothelial cells taken from bovine brains [39] and parts of brains in mice [38], rats and humans [40]. These insights support the presumption that tricellulin is critically involved in maintaining the integrity of the brain’s vasculature. Further, tricellulin mRNA was detected not only in association with the vasculature but also with neurons and astrocytes [40], which leads to the hypothesis that tricellulin may also play a role in maintaining the integrity of non-vascular elements within the NVU. Regarding the adjacent extracellular matrix, the present study revealed, at least in part, a regional association of tricellulin and the matrix-representing fibronectin in ischemia-affected brain areas. Considering the earlier observation of significantly altered immunosignals of collagen IV and fibronectin in areas of focal cerebral ischemia in mice [14], the regional association of tricellulin and fibronectin in affected brain areas seems plausible. For MFAP5, however, the immunofluorescence-based detection used in this study did not show a certain regional association with collagen IV and fibronectin as representatives of the vasculature and the extracellular matrix, neither in non-ischemic nor ischemic areas. Notably, ischemic areas were also characterized by a decreasing appearance of the originally fiber-like MFAP5 signal. Because MFAP5 has not yet been studied in the brain, a comparison of the present findings with those of other studies cannot be made. Although the immunosignal of α-catenin appeared mainly in a dotted pattern diminishing toward the ischemic region, a few regional associations with the immunosignals originating from collagen IV and fibronectin were seen. This observation seems plausible, as a very recent study with cell culture experiments was able to link tricellulin with α-catenin, which also displayed a dotted pattern [24]. In the setting of ischemia, the present observations suggest that both tricellulin and α-catenin provide a similar reaction. The ischemia-associated decreasing appearance of the α-catenin immunosignal seen in this study is further strengthened by an earlier observation, where a decreased signal was seen for β-catenin in the ischemic neocortices of mice [41]. 

To explore ischemia-related changes in tricellulin, MFAP5 and α-catenin concomitantly occurring with those of the cytoskeletal elements MAP2 and NF-L, immunosignals were quantified along the ischemic neocortex, including a border zone between the non- and the ischemia-affected region and in the subcortex. Regarding MAP2, the observed changes along the ischemia-affected neocortex were in line with a previous immunohistochemical study in mice, which showed a markedly decreasing signal of MAP2 toward the ischemic region [19]. For NF-L, the present study showed an increasing NF-L signal between the non-affected area and the ischemic border zone of the neocortex, whereas toward the ischemic region, a significantly lowered signal was found. A previous quantification of NF-L in the ischemia-affected neocortex of mice showed an increased NF-L signal towards ischemia, maintaining at an increased level within more laterally located ischemic regions [18]. Further quantification of the NF-L immunosignal in the ischemia-affected neocortices of mice revealed a regionally increased signal at the ischemic border zone with a subsequently decreasing signal in more lateral regions of the ischemic hemisphere [19]. Although the reported regionally increased NF-L signal at the border zone of ischemia is thus comparable with earlier observations, minor differences compared to earlier findings might have resulted from a slightly different allocation of the border zone within the affected neocortex and different immunoreagents. Along with an altered immunosignal of the cytoskeletal elements MAP2 and NF-L in the ischemia-affected neocortex, the present study revealed a gradually decreasing signal of tricellulin, MFAP5 and α-catenin toward the ischemic region of the neocortex, and also in the ischemic subcortex. However, for tricellulin, the decrease in the signal in the neocortical areas failed statistical significance after the consideration of multiple inter-hemispheric comparisons, but a significantly decreased signal was found comparing the non-affected neocortical region with the most injured neocortical region. Remarkably, for MFAP5 and α-catenin, different starting points of the decreased signal were found referring to the ischemic border zone and the cytoskeletal elements MAP2 and NF-L. While the MAP2 signal began to decrease in the medial part of the border zone, NF-L markedly decreased in the middle or lateral part of the border zone, which is a finding that is comparable with a previous investigation of these markers in the ischemia-affected brains of mice [19]. Interestingly, the MFAP5 signal decreased very early with respect to the ischemic lesion, i.e., close to the ischemic border zone of the neocortex, and remained at a decreased level along the ischemic area investigated. In contrast, the α-catenin signal started to decrease in the lateral part of the ischemic border zone of the neocortex and maintained at a decreased level. Regarding the local arrangement of tricellulin, MFAP5, α-catenin and the cytoskeletal elements MAP2 and NF-L, partially overlapping immunosignals were observed for the cytoskeletal elements with tricellulin and especially for MFAP5. Thereby, a fiber-like appearance became visible, particularly for MFAP5. In a subset of qualitative immunohistochemical analyses, the characteristics of signals from MAP2 and NF-L, as well as tricellulin, MFAP5 and α-catenin, were confirmed in mice and rats subjected to 4 h of focal cerebral ischemia. 

Although these immunohistochemical findings are naturally descriptive and a causal relationship remains speculative, the concomitantly observed ischemia-related alterations in vascular, extracellular and cytoskeletal elements, along with tricellulin, α-catenin and especially MFAP5, support the presumption that these elements may ensure cellular stabilization in the brain and may thus have a crucial role in the pathophysiology of stroke. These findings also support the concept of the NVU in the sense that ischemia affects not only neurons but also regionally associated cell types, including those forming the vasculature [6,7,8]. Furthermore, these findings may extend the NVU perspective toward the involvement of closely associated extracellular matrix constituents in the setting of ischemia, which was shown for fibronectin in a previous investigation [14] and for MFAP5 in the present study. Even though the family of MFAPs has been traditionally discussed to have a role in tissues with a high cellular turnover, e.g., in cancer or wound healing (overview in [42]), a link to the brain cytoskeleton might be mediated by integrins. This presumption is strengthened by previous cell culture experiments indicating that MFAP2 may bind via α_V_ß_3_ integrin to non-neuronal cell surfaces [43] and by the comprehensive roles of integrins that were discussed for the endothelium under ischemic conditions [44]. 

The present study has some limitations: First, immunofluorescence labeling was applied to detect tricellulin, MFAP and α-catenin in the non-ischemic and ischemic brain regions of mice and rats. This technique was chosen because it has already proven beneficial in visualizing elements of the NVU, the extracellular matrix and parts of the cytoskeleton, i.e., MAP2 and NF-L (e.g., [9,14,19]). Immunofluorescence labeling was successful in terms of a screening approach, first, to visualize tricellulin, MFAP and α-catenin in the rodent brain, and second, to explore regional associations with NVU and cytoskeletal elements. However, this study cannot provide details regarding protein levels and the functional relevance of observed signal alterations. Consequently, future studies should include techniques allowing the best possible detection of protein levels in specific brain areas, e.g., via Western blot analyses circumscribed to these areas. This seems important, as a recent study demonstrated that either increasing or decreasing the immunosignal does not necessarily correspond with changed protein levels in the same direction [45]. Moreover, future studies need to be focused on the functional relevance of ischemia-related alterations in tricellulin, MFAP5 and α-catenin, for example in respective knock-out animals. Second, regarding regional associations with cellular, extracellular and cytoskeletal elements, this study is based on immunosignals appearing partially overlapped and thus indicating a close regional association or even a co-localization, whose differentiation remains challenging. Consequently, future investigations are needed to dissolve this uncertainty while considering diverse markers for neuronal, glial and vascular cell types. Third, efforts were made to consider a second ischemia duration, i.e., 4 h in addition to 24 h, and a second species, i.e., the rat in addition to the mouse. However, stroke is characterized by many pathophysiological mechanisms of peaking at different time points after ischemia onset [1]. Future work should therefore include additional periods of ischemia along with an incomplete or even complete vessel recanalization to further explore the behavior of especially tricellulin and MFAP5 under these conditions. 

Collectively, this is the first study showing tricellulin, MFAP5 and α-catenin along with vascular, extracellular and cytoskeletal elements in non-ischemic and ischemic areas of the rodent brain. Whereas the immunosignals of tricellulin, MFAP5 and α-catenin mainly appeared with a diffusely arranged pattern, the tricellulin and notably MFAP5 signals partially appeared in a fiber-like pattern, and that of α-catenin appeared in a more dotted pattern. Regional associations with vascular and extracellular elements were partly seen for tricellulin and α-catenin, particularly in regions of focal cerebral ischemia. Along the ischemic neocortex, immunosignals of tricellulin, MFAP5 and α-catenin gradually decreased toward the most affected area concomitantly with altered signals of cytoskeletal elements. Compared with MAP2 and NF-L, MFAP5 was identified as most sensitive, as its signal changed very early with reference to the ischemic lesion. Although these findings need to be confirmed by further studies, the present observations suggest that tricellulin, α-catenin and notably MFAP5 may play a role as cell-stabilizing elements in the brain and may thus represent a target for neuroprotective approaches in stroke.

## 4. Materials and Methods

### 4.1. Study Design 

Applying an exploratory approach, this study used brains from mice and rats subjected to focal cerebral ischemia to address changes in different vascular, extracellular and cytoskeletal elements as visualized by immunohistochemical techniques. Analyses were based on 10 male C57/BL/6J mice with a mean body weight of 25.4 g, bred by Charles River Laboratories (Sulzfeld, Germany), and 4 male Wister rats with a mean body weight of 289.6 g, also bred by Charles River Laboratories (Sulzfeld). Focal cerebral ischemia was induced as described below. For histochemical investigations, mice were sacrificed after 4 (*n* = 4) or 24 h (*n* = 6) of ischemia, and rats (*n* = 4) were sacrificed after 4 h, using isoflurane (Isofluran Baxter, Baxter, Unterschleißheim, Germany).

Animal experiments were performed in accordance with the European Union Directive 2010/63/EU and the German guideline for the care and use of laboratory animals. Reporting followed the ARRIVE criteria [46]. Experiments were approved by Landesdirektion Sachsen (Leipzig, Germany) as the local authority (reference number: TVV 02/17, date of approval: 23 August 2017).

### 4.2. Experimental Focal Cerebral Ischemia 

In mice, focal cerebral ischemia with an affection of the right-sided middle cerebral artery territory was induced via a filament-based model, as described originally by Longa et al. [47] with minor modifications. In brief, followed by a medial cervical skin incision, right-sided cervical arteries were carefully exposed using a microscope (Carl Zeiss, Oberkochen, Germany). A silicon-coated 6-0 monofilament (Doccol Corporation, Redlands, CA, USA) was inserted into the internal carotid artery and moved forward until bending was observed or resistance was felt. The filament was left in place, and the skin was closed with a surgical suture. 

In rats, focal cerebral ischemia, also affecting the right-sided middle cerebral artery territory, was induced via a thromboembolic model described by Zhang et al. [48] with minor modifications. Briefly, right-sided cervical arteries were carefully exposed while using a microscope (Carl Zeiss). A polyethylene (PE) tube with a tapered end was introduced into the external carotid artery and moved forward through the internal carotid artery. At this position, a clot originating from rat blood was injected via the catheter, which was then removed. The skin was closed with a surgical suture. 

Surgical procedures were generally conducted under anesthesia using about 2–2.5% isoflurane (Isofluran Baxter, Baxter, Unterschleißheim, Germany) in a mixture of 70% 493 N_2_O/30% O_2_, applicated with a vaporizer (VIP 3000, Matrix, New York, NY, USA). For the prevention of anesthesia-associated cooling, a thermostatically controlled warming pad with a rectal probe (Fine Science Tools, Heidelberg, Germany) was used during surgical procedures, and 37 °C was adjusted as the target temperature. Mice and rats received a complex pain medication including lidocaine (Licain, DeltaSelect, Dreieich, Germany), meloxicam (Metacam, Boehringer Ingelheim Vetmedica, Ingelheim, Germany) and metamizole (Novaminsulfon-ratiopharm, Ratiopharm, Ulm, Germany), respectively. 

The sufficient induction of focal cerebral ischemia was verified clinically by at least a score of two using Menzies scoring [49] during the observation period after surgery, representing a pre-defined study inclusion criterion. This score naturally ranges from 0, indicating no deficit, to 4, indicating spontaneous contralateral circling. Included mice presented a mean Menzies score of 3.3, and that of rats was 3.0.

### 4.3. Tissue Preparation and Immunofluorescence Labeling 

After transcardial perfusion with saline and 4% phosphate-buffered paraformaldehyde (PFA), fixed brains from mice and rats were equilibrated with 30% phosphate-buffered sucrose. Forebrains were cut with a freezing microtome (Leica SM 2000R, Leica Biosystems, Wetzlar, Germany), and 10 series of 30 µm-thick sections were collected. Until further processing, brain sections were stored at 4 °C in 0.1 M Tris-buffered saline, with a pH of 7.4 (TBS), containing sodium azide. Immunofluorescence labeling started with rinsing sections in TBS followed by a one-hour antigen retrieval treatment with 60% methanol in water. For blocking non-specific binding sites, 5% normal donkey serum in TBS containing 0.3% Triton X-100 was applied for one hour. Next, sections were processed with different primary and secondary antibodies, as described in Table 1. Thereby, the green fluorescence signal of carbocyanine (Cy)2, the red signal of Cy3 and the blue (color-coded) signal of Cy5 visualized markers by means of a fluorescence-sensitive microscope or camera. 

In control experiments addressing the validity of visualized structures via immunofluorescence labeling, the primary antibodies were omitted, which resulted in the absence of specific signals.

### 4.4. Microscopy and Quantification of Fluorescence Signals

Brain sections, immunochemically stained with Cyanine-labeled secondary antibodies, were screened with an AxioScan.Z1 microscope (Carl Zeiss, Göttingen, Germany) equipped with a Colibri7 light source. Images of selected brain regions were taken with the Axiocam 506 (Carl Zeiss) and a Plan-Apochromat objective (20x/0.8; Carl Zeiss) with an exposure time of 7 ms for Cy2, 10 ms for Cy3, 5 ms for Cy5 and 2 ms for Hoechst 33342. Images were digitized using the ZEN 2.6 software (Carl Zeiss), and quantifications on immunosignals of tricelluin, MFAP5 and α-catenin were performed with ZEN 3.7 Intellesis software (Carl Zeiss) by measuring fluorescence intensities within defined regions of interests (ROIs). 

Therefore, 6 ROIs were arranged along the hemisphere that was affected by focal cerebral ischemia, and the third and fourth ROIs were located directly within the ischemic border zone. These 6 ROIs were mirrored to the contralateral hemisphere, which was naturally not affected by ischemia. In addition, a single ROI was placed in the ischemic subcortex and mirrored to the contralateral hemisphere. Each ROI had dimensions of 150 × 600 µm, and the squares were placed with a 50 µm gap within the neocortex. Quantifications were based on three brain sections per animal, and mean values of fluorescence intensities from each ROI were calculated across the three sections and used for intra- and interhemispheric comparison between ROIs. In case a single ROI failed to provide a valid fluorescence signal, e.g., due to mechanical alterations in the tissue, this ROI was not included in further processing.

### 4.5. Statistical Analyses and Figure Preparation

Based on immunosignals of tricellulin, MFAP5 and α-catenin, descriptive analyses and tests concerning statistical significance between regions were performed with the statistics software SigmaPlot 14.0 (Systat, Düsseldorf, Germany). Due to the relatively small sample size, a non-parametric test, i.e., the Wilcoxon Rank-Sum test, was applied. The Bonferroni–Holm correction was used to consider multiple comparisons and, thus, to reduce the risk of false-positive results [50]. Generally, a *p*-value of <0.05 was considered statistically significant. 

Figures comprising selected micrographs and a summary of quantifications were arranged with CorelDraw 24 (Corel Corporation, Ottawa, ON, Canada) and Microsoft PowerPoint (Office 365, Version 2023; Microsoft Corp., Redmond, WA, USA), and the brightness and contrast of micrographs were slightly adjusted if necessary but without deleting or creating signals. 

## Figures and Tables

**Figure 1 ijms-24-11893-f001:**
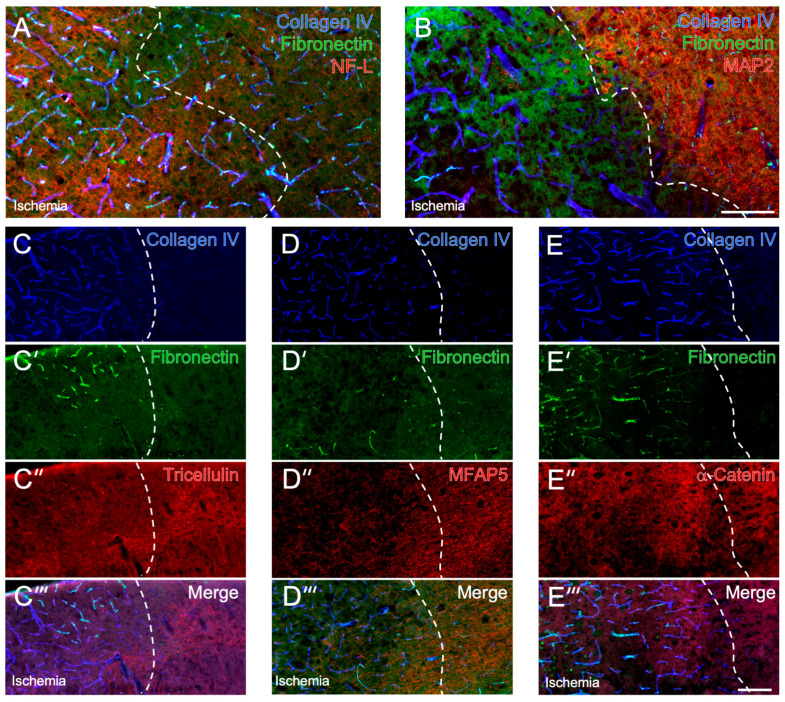
Immunohistochemical detection of vascular, extracellular and cytoskeletal elements along with tricellulin, α-catenin and microfibrillar-associated protein 5 (MFAP5) in mice subjected to 24 h of focal cerebral ischemia. Representative, multiple fluorescence labeling of collagen IV and fibro-nectin, representing elements of the vasculature and the extracellular matrix, combined with neurofilament light chain (NF-L, (**A**)) and microtubule-associated protein 2 (MAP2, (**B**)), both representing cytoskeletal elements, in the ischemia-affected neocortex. Triple immunofluorescence labeling of collagen IV and fibronectin combined with tricellulin (**C’’**), MFAP5 (**D’’**) and α-catenin (**E’’**), respectively. Scale bars: (**B**) (also valid for (**A**)): 100 µm, (**E’’’**) (also valid for (**C**–**C’’’**), (**D**–**D’’’**) and (**E**–**E’’**)): 100 µm.

**Figure 2 ijms-24-11893-f002:**
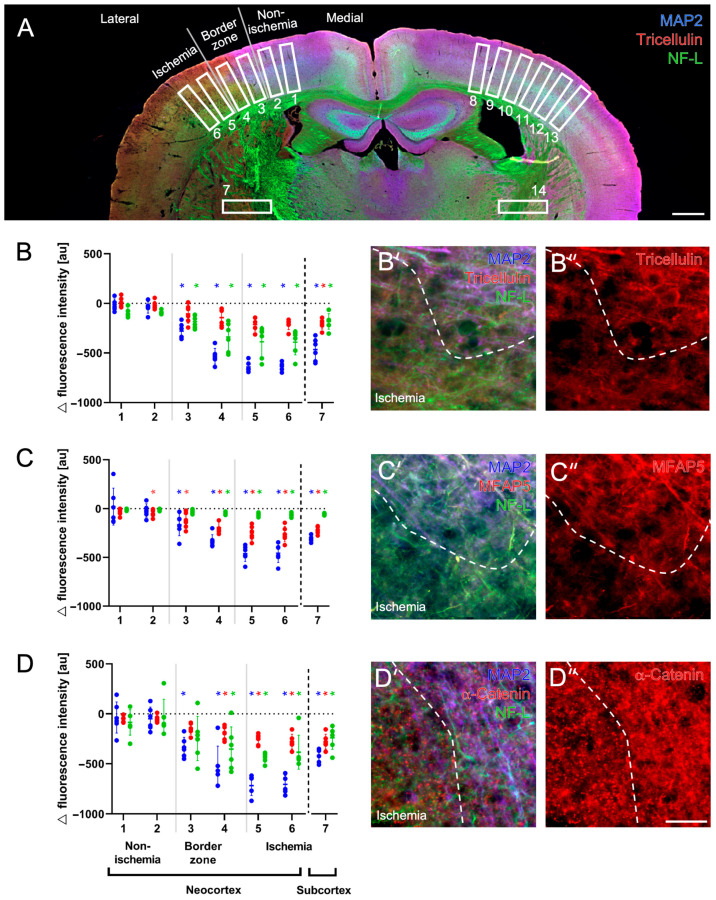
Quantifications of tricellulin, α-catenin and microfibrillar-associated protein 5 (MFAP5) along with the cytoskeletal elements of microtubule-associated protein 2 (MAP2) and neurofilament light chain (NF-L) in the ischemic neocortex and subcortex of mice subjected to 24 h of focal cerebral ischemia. Overview with exemplarily inserted regions of interest used for measurements of signal intensity with numbers 1 to 7 for the ischemia-affected hemisphere and numbers 8 to 14 for the non-affected hemisphere (**A**). Quantifications of respective immunosignals in terms of inter-hemispheric differences concerning the affected hemisphere (**B**–**D**), and values of each animal are shown as dots (blue: MAP2, green: NF-L, red: tricellulin, MFAP5 or α-catenin, respectively), added by vertical lines representing the standard deviation and horizontal marks crossing the lines representing the mean. Exemplary triple immunofluorescence labeling of MAP2 and NF-L combined with tricellulin (**B’**,**B’’**), MFAP5 (**C’**,**C’’**) and α-catenin (**D’**,**D’’**). Scale bars: (**A**): 500 µm, (**D’’**) (also valid for (**B’**,**B’’**,**C’**,**C’’**,**D’**): 25 µm. *: *p* < 0.05 (Wilcoxon Rank-Sum test with added Bonferroni–Holm correction for multiple comparisons).

**Figure 3 ijms-24-11893-f003:**
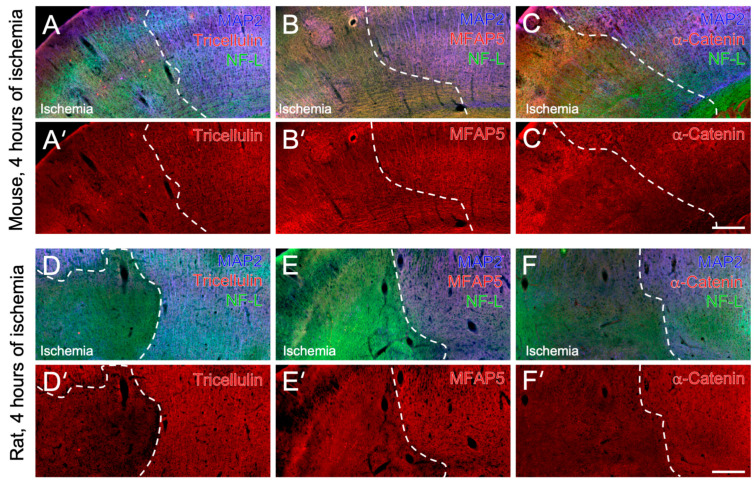
Immunohistochemical detection of cytoskeletal elements with tricellulin, α-catenin and microfibrillar-associated protein 5 (MFAP5) in mice and rats subjected to 4 h of focal cerebral ischemia. Representative triple immunofluorescence labeling of microtubule-associated protein 2 (MAP2) and neurofilament light chain (NF-L) combined with tricellulin (**A**,**A’**), MFAP5 (**B**,**B’**) and α-catenin (**C**,**C’**), respectively, in the mouse’s neocortex. In a similar formation, triple fluorescence labeling of MAP2 and NF-L combined with tricellulin (**D**,**D’**), MFAP5 (**E**,**E’**) and α-catenin (**E**,**E’**), respectively, in the rat’s neocortex. Scale bars: (**C’**) (also valid for (**A**,**A’**,**B**,**B’**,**C**)): 100 µm, (**F’**) (also valid for (**D**,**D’**,**E**,**E’**,**F**)): 100 µm.

**Table 1 ijms-24-11893-t001:** Overview of applied immunoreagents.

Antigen	Host	Dilution	Supplier	Product Number	Stock Concen-tration	Clone	Secondary Antibodies *
Fibronectin	chicken	1:100	Agrisera	IMS02-060-314	1 mg/mL	polyclonal	donkey anti-chicken Cy2 Dianova 703-225-155
Collagen IV	goat	1:40,000	Merck	AB769	0.4 mg/mL	polyclonal	donkey anti-goat Cy5 Dianova 705-175-147
NF-L	mouse	1:100	Life Technologies	130400	0.5 mg/mL	DA2 (IgG1k)	donkey anti-mouse Cy3 Dianova 715-165-151 donkey anti-mouse-Cy2 Dianova 715-225-150
MAP2	guinea pig	1:500	Synaptic Systems	188004	not available	polyclonal	donkey anti-guinea pig Cy3 Dianova 706-165-148 donkey anti-guinea pig Cy5 Dianova 706-175-148
MFAP5	rabbit	1:250	Sigma	HPA010553	1 mg/mL	polyclonal	donkey anti-rabbit Cy3 Dianova 711-165-152
Tricellulin	rabbit	1:100	Invitrogen	700191	0.5 mg/mL	54H19L38 (IgG)	donkey anti-rabbit Cy3 Dianova 711-165-152
α-catenin (biotin conjugated)	mouse	1:100	Antibodies -online	ABIN6178165	0.1 mg/mL	1G5 (IgG1)	Streptavidin Cy3 Dianova 016-160-084

* All fluorochromated secondary immunoreagents were from Jackson ImmunoResearch (West Grove, PA, USA).

## Data Availability

Data underlying this study will be made available upon reasonable request.

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
