# Peer review of "Tricellulin, α-Catenin and Microfibrillar-Associated Protein 5 Exhibit Concomitantly Altered Immunosignals along with Vascular, Extracellular and Cytoskeletal Elements after Experimental Focal Cerebral Ischemia"

_ijms, 2023, doi:10.3390/ijms241511893_

Round 1
Reviewer 1 Report
The authors studied the expression of proteins part of the extracellular matrix (ECM) following focal stroke. They are focusing on cell-stabilizing elements, with MAP2 (microtubule-associated protein-2), the neurofilament light chain (NF-L) and alpha-catenin proteins part of the cytoskeleton; tricellulin – transmembrane protein allowing contact between three adjoining cells, and MFAP5 (microfibrillar-associated proteins); extracellular matrix glycoproteins, collagen IV and fibronectin. They used the filament-based-model in rats and mice at 4 and 24hrs. Using immunofluorescence on frozen fixed brain sections, they showed that tricellulin, MFAP5, α-catenin, MAP2 and NF-L decreased in the ischemic zone.
While the study of the ECM related to the neurovascular unit is interesting, several flaws in the methods make this study of no real significant value.
The present study is exclusively based on fluorescent immunohistochemistry, and the authors do not present the controls necessary to validate their findings. No immunostaining of the studied proteins is presented in Naïve and Sham animals. At least two antibodies per protein of interest should have been used and the validation of each antibody should be provided. The authors cannot convincingly describe and conclude alterations in the protein expression detected without comparing with healthy tissue.
Qualitative description of the immunostaining is not sufficient, and quantification should be provided for all figures. Since their study is mostly descriptive, a 3-D rendering of the proteins studied related to the neurovascular unit and how it is altered after ischemia is expected.
Reviewer 2 Report
The manuscript by Höfling et al. shows for the first time the morphological characterization of Tricellulin, α-catenin, and microfibrillar-associated protein 5 in mice/rat models of focal cerebral ischemia.
The manuscript is descriptive and lacks functional aspects of this connection. The experimental point of view is uniquely by imaging perspective, but it could be considered the first step of the exploration of tricellulin, MFAP, and α-catenin in rodent models of focal cerebral ischemia in conjunction with vascular and extracellular constituents, and the cytoskeletal elements MAP2 and NF-L.
The experimental plan is well executed, and the data are clear and well-written. The literature cited is of important relevance. The methods description is correct, and obviously, in the discussion are evidenced many limits of these descriptive results, but this is not due to the authors but to the limited reports of the research theme. More molecular and cellular details are requested, but cellular extract fractions aimed to quantify (western blotting) the distribution of tricellulin, MFAP5, and α-catenin with cytoskeletal MAP2 and NF-L are important to validate these immunostaining observations.
Lastly, this study is interesting but an alternative quantification method to evaluate the protein expression is needed. Alternatively an in situ RNA hybridization could permit to evaluate a concomitant/or not concomitant local mRNA expression in ischemic tissues.
Acceptable only after major revision.
Reviewer 3 Report
Hoefling and co-authors studied the effects of cerebral ischemia in mice and rats on the expression of alpha-catenin, tricellulin and MFAP5 in brain. Using immunofluorescence microscopy, the authors found that the expression of all three proteins is decreased in the ischemic zones. This paper presents purely descriptive results, without an attempt to go deeper into the biological meaning of the observed effects. Moreover, it is not clear in which cell types of brain are expressed the studied proteins. Co-staining for specific markers of neurons, glial cells and endothelial cells would be helpful. In addition, the immunofluorescence data should be verified by comparative immunoblotting study of proteins expression in ischemic and control areas.
Round 2
Reviewer 2 Report
The reply comments to criticism are robustly discussed and suggest a clear interpretation of experimental limits, but also a sharp knowledge of the field. The shift to IJMS 'Communication' fit with the message.
The manuscript is accepted as is.
Reviewer 3 Report
The authors properly addressed the limitations of the study. The paper is acceptable as a Communication.